# The Effect of Acceptor Structure on Emission Color Tuning in Organic Semiconductors with D–π–A–π–D Structures

**DOI:** 10.3390/nano9081179

**Published:** 2019-08-17

**Authors:** Przemyslaw Ledwon, Gabriela Wiosna-Salyga, Marian Chapran, Radoslaw Motyka

**Affiliations:** 1Faculty of Chemistry, Silesian University of Technology, Strzody 9, 44-100 Gliwice, Poland; 2Department of Molecular Physics, Lodz University of Technology, Zeromskiego 116, 90-924 Lodz, Poland

**Keywords:** fluorescence, electron-acceptor units, charge transfer, near-infrared emission, organic light emitting diode (OLED)

## Abstract

A series of novel donor–acceptor D–π–A–π–D compounds were synthesized and characterized in order to determine the influence of different acceptor units on their properties. The introduction of acceptor moieties had a direct impact on the HOMO and LUMO energy levels. Fluorescence spectra of compounds can be changed by the choice of an appropriate acceptor and were shifted from the green to the near-infrared part of spectra. Due to observed concentration induced emission quenching, the green exciplex type host was used to evaluate the potential of synthesized molecules as emitters in organic light emitting diodes (OLEDs).

## 1. Introduction

Donor–acceptor compounds are an important group of π-conjugated organic materials. One of the main factors affecting the properties of such fluorescent compounds is their structure, which includes a donor, an acceptor, and a linker between the two. Some research has focused on the effects of different acceptor units on photophysical properties [1,2,3,4], but most studies have mainly focused on photovoltaic applications [5,6,7].

In recent years, there has been a growing interest in near-infrared (NIR) to deep-red emitters [8], because of their possible applications in telecommunication and night vision technologies. There is also an increasing demand for NIR emitters in sensing, in areas such as medicine [9], optical communication, lasers, and the automotive industry. This means that the market for these devices in the near future may be one of the most rapidly developed areas of research. In this respect, a study focusing on emission tuning to achieve deep-red and infrared emission is important. For this reason, the development of new emitters with an entire palette of emission colors is highly justified.

Emitters for organic light emitting diodes (OLEDs) can be divided into inorganic semiconductors, metal complexes, and organic compounds [10,11], and there is growing interest in completely OLEDs. Currently, most commercially-available OLEDs are mainly based on Ir or Pt emitters because of their high efficiency and working stability [12,13,14]. In such devices, the emission layer consists of the host organic material and guest phosphorescence dopant. The main drawback of such devices is the use of precious metals, which results in irreversible damage to the environment. One of the most promising solutions is the use of fully organic fluorescent emitters which exhibit thermally activated delayed fluorescence (TADF) [15,16,17]. In the typical fluorescence-based devices a neat film of the emitter or the host-guest system can be used as an emission layer [18]. In addition, a common fluorescent molecule combined with a TADF emitter can be used in hyperfluorescent OLEDs [19]. In such devices, a TADF emitter acts as a sensitizer that harvests excitons and transfers them to a fluorescent emitter by Förster energy transfer [20]. This could be an important strategy that will enable the fabrication of efficient and stable fully-organic OLEDs.

The latest reports indicate that purely fluorescent organic materials can be also efficient NIR emitters [21,22,23,24,25]. One of the most promising strategies used to obtain organic molecules with emission in the red and infrared range is the connection of a strong electron-acceptor (A) unit with a strong electron-donor (D) unit to form polymer or oligomer with structure D–A, D–A–D or A–D–A [26]. The structure can also be modified by adding a π-conjugated bridge (π) which can increase the effective conjugation length. The choice of the appropriate D, A, and π units is crucial to control over the material functionality. Carbazole is one of the most frequently used donor units in many D–A type of dyes, owning to its advantages, such as high luminescent efficiency and broad ability to modify its structure [27]; so far different electron-accepting moieties and their derivatives, including benzoselenadiazole [28], benzothiadiazole [29], quinoxaline [30], and thienopyrroledione [31] were reported as acceptor units with low LUMO levels. However, there are only a limited number of reports that analyze the impact of different types of strong acceptor units on luminescence properties [32].

In most cases, organic emitters are implemented in multilayer OLED structures fabricated by vacuum deposition, but this approach increases the production costs and restricts the implementation of NIR emitters [33,34,35]. An alternative to this is the fabrication of OLEDs by solution-based techniques, such as inkjet printing, screen printing, slot die coating, and spin coating. These methods reduce the production costs and can be used to produce large and flexible organic devices. It should be emphasized that to realize this, NIR emitters should be suitable for solution processing techniques. The corresponding approach can be used with donor–acceptor compounds with controlled energy band gaps and responsive emission.

Herein, we report the design, synthesis, and characterization of solution-processable organic semiconductors with D–π–A–π–D structures (Figure 1): 1,3-bis(5-((E)-2-(9-(2-ethylhexyl)-9H-carbazol-3-yl)vinyl)thiophen-2-yl)-5-octyl-4H-thieno[3,4-c]pyrrole-4,6(5H)-dione (**C1**), 5,8-bis(5-((E)-2-(9-(2-ethylhexyl)-9H-carbazol-3-yl)vinyl)thiophen-2-yl)quinoxaline (**C2**), 4,7-bis(5-((E)-2-(9-(2-ethylhexyl)-9H-carbazol-3-yl)vinyl)thiophen-2-yl)-5,6-difluorobenzo[c][1,2,5]thiadiazole (**C3**), 4,7-bis(5-((E)-2-(9-(2-ethylhexyl)-9H-carbazol-3-yl)vinyl)thiophen-2-yl)benzo[c][1,2,5]selenadiazole (**C4**), and 5,5’-bis((E)-2-(9-(2-ethylhexyl)-9H-carbazol-3-yl)vinyl)-2,2’-bithiophene (**C5**). A new series of model molecules was designed and developed to examine the possibility of shifting emissions from the visible to the near-infrared region using common building blocks. Structures containing thiophene, vinyl, and carbazole units were chosen to ensure extensive π-conjugation as a result of the planarization of the vinyl linker, and ethylhexyl groups were used to ensure sufficient solubility of the final compounds. Different acceptor moieties were chosen from commercially-available compounds with low-lying LUMO levels. Following this, the effect of the acceptor moieties was studied by photophysics, and the molecules were later implemented in electroluminescence devices.

## 2. Materials and Methods

### 2.1. Synthesis

^1^H- and ^13^C-NMR spectra were recorded on a Varian 300 MHz system (300 MHz for ^1^H and 75 MHz for ^13^C), in specified solvents using tetramethylsilane as the internal reference. The chemical shifts (δ) are reported in parts per million (ppm) and the coupling constants (*J*) in Hertz. HRMS analysis were performed on a Waters Xevo G2 Q-TOF mass spectrometer (Waters Corporation, MA, USA) with electrospray ionization. Anhydrous 2-ethylhexylamine (97%), DMSO, and *n*-butyllithium (2.5 M in hexane) were purchased from ACROS Organics (Thermo Fisher Scientific, Geel, Belgium) and used as obtained. Dichloromethane (DCM, for HPLC, 99.8%) and Pd(PPh_3_)_4_ were obtained from Sigma-Aldrich (Saint Louis, MO, USA), and hexanes fractionated from petroleum were obtained from Avantor Performance Materials Poland S.A. (Gliwice, Poland).

**9-(2-ethylhexyl)-9*H*-carbazole** (**2**): A modified procedure for the alkylation of phenols was modified from R. Motyka et al. [36]. 9*H*-carbazole (**1**) (5.0 g, 30.0 mmol) and powdered potassium hydroxide (1.85 g, 33.0 mmol) were mixed in DMSO (50 mL) under an argon atmosphere for 10 min. Then, 2-ethylhexyl bromide (97%) (5.95 g, 5.48 mL, 30.0 mmol) was added dropwise with stirring, and stirring was continued for 12 h. Water (300 mL) was added to the reaction mixture, which was then extracted with DCM (3 × 50 mL). The extracts were combined and washed with water and brine and then dried over anhydrous MgSO_4_. The resulting solution was evaporated, and the residue was purified using silica gel column chromatography with hexane (fractionated from petroleum 65–80 °C) as an eluent to yield 7.50 g (90% yield) of product (**2**) as colorless thick oil. The structure of the compound was confirmed by NMR analysis [37]. **^1^H-NMR** (300 MHz, CDCl_3_) δ (ppm): 8.10 (m, 2H, C_4,5Ar_–H), 7.51–7.42 (m, 2H, C_2,7Ar_–H), 7.39 (d, ^3^*J* = 7.7 Hz, 2H, C_1,8Ar_–H), 7.29–7.17 (m, 2H, C_3,6Ar_–H), 4.35–3.97 (m, 2H, >N–CH_2_–), 2.16–1.98 (m, 1H, –CH<), 1.50–1.28 (m, 8H, 4 × –CH_2_–), and 1.00–0.75 (m, 6H, 2 × –CH_3_).

**9-(2-ethylhexyl)-9*H*-carbazole-3-carbaldehyde** (**3**): Using a procedure adapted from Z. Yang et al. [38], 9-(2-ethylhexyl)-9*H*-carbazole (**2**) (7.40 g, 26.4 mmol), DMF (1.93 g, 2.05 mL, 26.4 mmol) and 30 mL of anhydrous CHCl_3_ were placed into a dried, 50 mL round-bottom flask equipped with a magnetic stirrer and reflux condenser. The entire system was flushed with inert gas and cooled to 0 °C with vigorous stirring. To this mixture, POCl_3_ (4.05 g, 2.50 mL) was added dropwise via syringe, and afterwards the cooling bath was removed, and the mixture was heated under reflux for 12 h. Next, most CHCl_3_ was evaporated, and the residue was poured into ice water (100 mL), and the pH was adjusted to 7–8 by adding NaHCO_3_. The aqueous suspension was extracted with CH_2_Cl_2_ (3 × 25 mL), and the combined extracts were washed with water several times before being dried with anhydrous MgSO_4_. After CH_2_Cl_2_ was removed, the crude product was purified by silica gel column chromatography with hexane/DCM (2:1, *v*/*v*) as an eluent to afford 5.50 g (67% yield) of a pale-yellow liquid product (**3**). The structure of the compound was confirmed by NMR analysis [37]. **^1^H-NMR** (300 MHz, CDCl_3_) δ(ppm): 10.09 (s, 1H, –CHO), 8.60 (d, ^4^*J* = 1.6 Hz, 1H, C_4Ar_–H), 8.26–8.07 (m, 1H, C_5Ar_–H), 8.00 (dd, ^3^*J* = 8.6 Hz, ^4^*J* = 1.6 Hz, 1H, C_2Ar_–H), 7.53 (ddd, ^3^*J* = 8.3 Hz, ^3^*J* = 7.1 Hz, ^4^*J* = 1.2 Hz, 1H, C_Ar_–H), 7.48–7.40 (m, 2H, 2 × C_Ar_–H), 7.32 (ddd, ^3^*J* = 8.0 Hz, ^3^*J* = 7.1 Hz, ^4^*J* = 1.0 Hz, 1H, C_Ar_–H), 4.39–3.91 (m, 2H, >N–CH_2_-), 2.24–1.86 (sept., ^3^*J* = 6.0 Hz, 1H, –CH<), 1.54–1.08 (m, 8 H, 4 × –CH_2_–), 0.92 (*t*, ^3^*J* = 7.4 Hz, 3H, –CH_3_), and 0.85 (*t*, ^3^*J* = 7.1 Hz, 3H, –CH_3_). **^13^C-NMR** (75 MHz, CDCl_3_) δ(ppm): 191.86 (–CHO), 144.68, 141.76, 128.61, 127.22, 126.79, 124.05, 123.15, 123.10, 120.80, 120.40, 109.83, 109.36, 47.85 (>N–CH_2_–), 39.51, 31.12, 28.89, 24.52, 23.13, 14.12,and 11.02.

**(9-(2-ethylhexyl)-9*H*-carbazol-3-yl)methanol** (**4**): The synthesis was performed according to a modified procedure described by D. Barpuzary et al. [39]: 3-formyl-*N*-(2-ethylhexyl)carbazole (**3**) (3.92 g, 12.75 mmol) and 50 mL of dry THF were placed in a 100 mL round-bottom flask, which was purged with argon, and the contents were cooled to 0 °C. Then, with vigorous stirring, sodium borohydride (0.53 g, 14.0 mmol) was added to the mixture in small portions, and stirring was continued overnight at room temperature. Afterwards, the reaction mixture was poured into cold water and extracted several times with dichloromethane. The combined organic layers were dried using anhydrous magnesium sulphate, and the solvent was removed under vacuum. The crude product was purified by silica column chromatography using dichloromethane as an eluent to yield 3.27 g (83% yield) of (**4**) as a clear oil. **^1^H-NMR** (300 MHz, CDCl_3_) δ(ppm): 8.16–7.96 (m, 2H, C_Ar_–H), 7.51–7.41 (m, 2H, C_Ar_–H), 7.41–7.29 (m, 2H, C_Ar_–H), 7.27–7.16 (m, 1H, C_Ar_–H), 4.81 (s, 2H, –CH_2_–OH), 4.12 (d, ^3^*J* = 7.0 Hz, 2H, >N–CH_2_-), 2.12–1.96 (m, 1H, –CH<), 1.87 (s, 1H, –OH), 1.50–1.11 (m, 8 H, –CH_2_-), and 0.99–0.71 (m, 6 H, 2 × –CH_3_).

**[(9-(2-ethylhexyl)-9*H*-carbazol-3-yl)methyl]triphenylphosphonium bromide (5)**: The synthesis was performed according to a modified procedure described by D. Barpuzary et al. [39]. A 100 mL round-bottom flask with a condenser was charged with compound **4** (3.25 g, 10.5 mmol), triphenylphosphine hydrobromide (3.60 g, 10.5 mmol), and 60 mL of DCM. The mixture was heated under reflux under an inert atmosphere for 12 h. After cooling to room temperature, the product was precipitated with diethyl ether, but the precipitate was a very thick viscous liquid. Neither rubbing with a glass rod, nor precipitation using benzene instead of Et_2_O yielded results. Therefore, solvent was decanted from the sticky product, and the residue, after additional evaporation and under reduced pressure (6.0 g, 90% yield) was used directly in the next reaction without further purification.

**(*E*) & (*Z*)-3-(2-(5-bromothiophen-2-yl)vinyl)-9-(2-ethylhexyl)-9*H*-carbazole** (**6**): This synthesis was performed based on a modified procedure from our previous publication, P. Ledwoń et al. [40] 5-bromo-2-formylthiophene (1.80 g, 9.45 mmol), compound **5** (6.00 g, 9.45 mmol), and a catalytic amount of 18-crown-6 ether (0.40 g, 1.50 mmol) were dissolved in dry methylene chloride (300 mL). Then, anhydrous potassium carbonate (4.00 g, 30.0 mmol) was added. The resulting slurry was intensely stirred and heated under reflux for 18 h under argon, then cooled, filtered, and the solvent was removed under reduced pressure. The remaining viscous brown oil was purified using silica gel chromatography, using a mixture of dichloromethane (DCM) and hexane (1:4 *v*/*v*) as the eluent. Column chromatography gave one main fraction containing both isomers, which after evaporation gave 3.9 g (8.4 mmol, 90% yield) of a pale yellow solid. The obtained mixture was not separated into individual components. From the ^1^H-NMR spectrum analysis, it was possible to determine that the obtained mixture contained an almost equimolar amount of each isomer. **^1^H-NMR** (300 MHz, CDCl_3_) δ(ppm): 8.20–8.00 (m, 2H, C_Ar_–H), 7.57 (dd, ^3^*J* = 8.6 Hz, ^4^*J* = 1.6 Hz, 0.5H, C_Ar_–H), 7.52–7.29 (m, 4H, C_Ar_–H), 7.29–7.19 (m, 1H, C_Ar_–H), 7.14 (d, ^3^*J* = 15.9 Hz, 0.5H, H_vinyl_-isomer*E*), 7.01 (d, ^3^*J* = 15.9 Hz, 0.5H, H_vinyl_-isomer*E*), 6.95 (d, ^3^*J* = 3.8 Hz, 0.5H, C_Th_–H), 6.87–6.71 (m, 1H, C_Ar_–H), 6.61 (d, ^3^*J* = 11.7 Hz, 0.5H, H_vinyl_-isomer*Z*), 4.32–3.98 (m, 2H, >N–CH_2_–), 2.16–1.97 (m, 1H, –CH<), 1.48–1.17 (m, 8H, –CH_2_–), 1.00–0.80 (m, 6H, –CH_3_).

**Stannylation procedure**: A mixture of **6** (1.00 g, 2.14 mmol) in dry THF (15 mL) was cooled to −78 °C, then *n*-butyllithium (1.10 mL, 2.5 M in hexanes, 2.75 mmol) was added dropwise over a period of 1 h. After complete addition, the mixture was stirred at −78 °C for 2 h, and tributyltin chloride (95%) (0.73 g, 2.75 mmol) was added dropwise over a period of 15 min. The mixture was allowed to warm to room temperature and was stirred overnight. Then, the flask’s contents were poured into a mixture of DCM and saturated ammonium chloride solution in H_2_O (200 mL, 1:1 *v*/*v*) and vigorously shaken for 15 min. The organic layer was separated and dried using anhydrous MgSO_4_. After filtration, solvents were removed in vacuo to give 1.75 g of crude product (**7**) in the form of a light brown oil that quickly darkened in air. NMR analysis showed that the resulting oil contained a small amount of THF and around 80% of targeted compounds. Due to its very low stability, the crude product was used without further purification.

**Stille reaction—main procedure**: The appropriate acceptor (0.45 mmol) and toluene (10 mL) were placed in a dried 25 mL round-bottom flask equipped with a reflux condenser and inert gas inlet. The contents of the flask were vigorously stirred and purged with argon for 0.5 h. After this time, 0.79 g (1.20 mmol) of a mixture of **7** in 4 mL of degassed toluene was injected, followed by 25.0 mg (22 µmol, 5%) of Pd(PPh_3_)_4_ and 8.0 mg (44 µmol, 10%) of copper(I) iodide (CuI). The reaction mixture was heated under reflux under an inert atmosphere for 24 h. After cooling, the contents of the flask were poured into a mixture of DCM and a saturated aqueous solution of KF (200 mL, 1:1 *v*/*v*) and vigorously stirred for 1 h. The organic layer was separated, washed twice with deionized water (25 mL), and dried over anhydrous magnesium sulphate. After evaporation, the residue was purified using silica gel chromatography with a mixture of chloroform/hexane (fraction from petroleum, 65–80 °C), 2:1 (*v*/*v*), as the eluent to give a mixture of final compounds as isomers (**C1**, **C2**, **C3**, **C4**). **Isomerization**: 200 mg of the isomer mixture of each compound was dissolved in DCM (10 mL), to which trifluoroacetic acid (5 mL) was added, followed by water (2 mL). The resulting dark mixture was stirred vigorously for 30 min, after which the organic layer was separated, washed with water and diluted ammonia, and dried over magnesium sulphate. After evaporation of DCM, the resulting solid was purified by silica gel column chromatography using chloroform/hexane 2:1 (*v*/*v*) as the eluent. After solvent evaporation, the solid was dissolved in a minimal amount of DCM and was precipitated in methanol to give the product in the form of the most thermodynamically-stable isomer, EE.

**1,3-bis(5-((*E*)-2-(9-(2-ethylhexyl)-9*H*-carbazol-3-yl)vinyl)thiophen-2-yl)-5-octyl-4*H*-thieno-[3,4-*c*]-pyrrole-4,6(5*H*)-dione (C1)**: Substrate, 1,3-dibromo-5-octyl-4*H*-thieno-[3¨C-*c*]-pyrrole-4,6(5*H*)-dione 0.10 g (0.24 mmol), reaction gave 225 mg of a mixture of three isomers, from which 195 mg (92% yield) of the desired product was obtained as a brown powder. Melting point: 79–81 °C. **^1^H-NMR** (300 MHz, CD_2_Cl_2_) δ(ppm): 8.13 (s, 2H, Carb–H4), 8.08 (d, ^3^*J* = 7.6 Hz, 2H, Carb–H5), 7.89 (d, ^3^*J* = 3.9 Hz, 2H, Th–H3), 7.57 (dd, ^3^*J* = 8.6 Hz, ^4^*J* = 1.4 Hz, 2H, Carb–H2), 7.52–7.42 (m, 2H, Carb–H6), 7.42–7.28 (m, 4H, Carb–H1,8), 7.28–7.20 (m, 2H, Carb–H7), 7.17 (s, 4H, Vinyl-H1,H2), 7.01 (d, ^3^*J* = 3.9 Hz, 2H, Th-H4), 4.08 (d, ^3^*J* = 7.1 Hz, 4H, >N–CH_2_–), 3.58 (t, ^3^*J* = 7.4 Hz, 2H, >N–CH_2_–), 2.18–1.90 (m, 2H, –CH<), 1.78–1.60 (m, 2H, –CH_2_–), 1.49–1.12 (m, 26H, –CH_2_–), and 1.03–0.76 (m, 15H, – CH_3_). **^13^C-NMR** (75 MHz, CD_2_Cl_2_) δ(ppm): 162.75, 147.49, 141.75, 141.37, 136.10, 131.84, 131.02, 130.62, 128.58, 127.92, 126.73, 126.24, 124.78, 123.51, 123.04, 120.61, 119.46, 119.04, 118.53, 109.75, 109.69, 47.84, 39.78, 38.77, 32.22, 31.37, 29.62, 29.16, 28.90, 27.40, 24.78, 23.42, 23.04, 14.24, 14.14, and 11.04. **MALDI-TOF MS** calculated mass for C_66_H_74_N_3_O_2_S_3_+ [M+H]^+^: 1036 (100%), 1037 (76%), and 1039 (42%) found: 1036 (100%), 1037 (77%), and 1039 (40%).

**5,8-bis(5-((*E*)-2-(9-(2-ethylhexyl)-9*H*-carbazol-3-yl)vinyl)thiophen-2-yl)quinoxaline (C2)**: The substrate–5,8-dibromoquinaxoline 0.15 g (0.52 mmol), reaction gave 427 mg of a mixture of isomers, from which 365 mg (78% yield) of the desired product was obtained as a brown powder. Melting point: 135–137 °C. **^1^H-NMR** (300 MHz, CDCl_3_) δ(ppm): 9.01 (s, 2H, Quin–H2,H3), 8.22 (d, *J* = 1.4 Hz, 2H, Quin–H6,H7), 8.18–8.06 (m, 4H, Carb–H4,H5), 7.77 (d, ^3^*J* = 3.9 Hz, 2H, Th-H3), 7.65 (dd, ^3^*J* = 8.6 Hz, *^4^J* = 1.6 Hz, 2H, Carb–H2), 7.55–7.43 (m, 2H, Carb–H6), 7.43–7.33 (m, 4H, Carb–H1,8), 7.31 (s, 4H, Vinyl–H1,H2), 7.29–7.20 (m, 2H, Carb–H7), 7.14 (d, *^3^J* = 3.9 Hz, 2H, Th-H4), 4.15 (d, *^3^J* = 7.0 Hz, 4H, >N–CH_2_–), 2.22–1.92 (m, 2H, –CH<), 1.52–1.16 (m, 16H, –CH_2_-), 0.92 (*t*, *^3^J* = 7.4 Hz, 6H, –CH_3_), and 0.87 (*t*, *^3^J* = 7.0 Hz, 6H, –CH_3_). **^13^C-NMR** (75 MHz, CDCl_3_) δ(ppm): 146.73, 143.37, 141.45, 140.87, 139.96, 136.58, 131.77, 129.85, 128.33, 127.83, 127.23, 125.92, 125.49, 124.39, 123.33, 122.94, 120.50, 119.55, 119.15, 118.70, 109.37, 109.31, 47.65, 39.56, 31.14, 28.95, 24.53, 23.19, 14.18, and 11.05. **HRMS** (ESI) calculated mass for C_60_H_60_N_4_S_2_^+^ [M]^+^: 900.4259, found: 900.4296; C_60_H_61_N_4_S_2_^+^ [M+H]^+^: 901.4332 (100.0%), 902.4366 (64.9%), and 903.4399 (20.7%), found: 901.4333 (100%), 902.4351 (50%), and 903.4368 (30%).

**4,7-bis(5-((*E*)-2-(9-(2-ethylhexyl)-9*H*-carbazol-3-yl)vinyl)thiophen-2-yl)-5,6-difluorobenzo[*c*]-[1,2,5]-thiadiazole (C3)**: Substrate–4,7-dibromo-5,6-difluorobenzo[1,2,5]thiadiazole 0.15 g (0.45 mmol), reaction gave 320 mg of a mixture of isomers, from which 280 mg (66% yield) of desired product was obtained as a dark brown powder. Melting point: 237–239 °C. **^1^H-NMR** (300 MHz, CDCl_3_) δ(ppm): 8.31–8.18 (m, 4H, Carb–H4, Th–H3), 8.13 (d, *^3^J* = 7.6 Hz, 2H, Carb–H5), 7.66 (dd, *^3^J* = 8.6 Hz, *^4^J* = 1. 3 Hz, 2H, Carb–H2), 7.54–7.43 (m, 2H, Carb–H6), 7.43–7.23 (m, 10H, Carb–H1,7,8, Vinyl–H1,2), 7.21 (d, *^3^J* = 4.0 Hz, 2H, Th–H4), 4.16 (d, *^3^J* = 6.9 Hz, 4H, >N–CH_2_–), 2.16–1.96 (m, 2H, –CH<), 1.47–1.15 (m, 16H, –CH_2_–), and 1.01–0.76 (m, 12H, –CH_3_).**^13^C-NMR** (150 MHz, CDCl_3_) δ(ppm): 150.02, 149.07, 147.74, 147.18, 144.81, 141.51, 131.92, 126.06, 125.87, 124.53, 123.38, 122.91, 120.54, 119.29, 118.91, 109.41, 109.37, 47.70, 39.59, 31.18, 28.99, 24.57, 23.20, 14.20, and 11.06. Not all of the carbon signals are visible in the spectrum due to the extremely low solubility of the compound and the presence of fluorine atoms in its structure, the coupling of which with the ^13^C carbon nuclei additionally splits and reduces the size of recorded signals. **HRMS** (ESI) calculated mass for C_58_H_56_F_2_N_4_S_3_^+^ [M]^+^: 942.3635, found: 942.3627; C_58_H_57_F_2_N_4_S_3_^+^ [M+H]^+^: 943.3708 (100.0%) and 944.3741 (62.7%), found: 943.3697 (100%) and 944.3669 (60%).

**4,7-bis(5-((*E*)-2-(9-(2-ethylhexyl)-9*H*-carbazol-3-yl)vinyl)thiophen-2-yl)benzo[*c*][1,2,5]selena-diazole (C4)**: Substrate–4,7-dibromobenzo[*c*][1,2,5]selenadiazole 0.15 g (0.44 mmol), reaction gave 220 mg of a mixture of three isomers, from which 190 mg (45% yield) of desired product was obtained as a dark blue powder. Melting point: 139–141 °C. **^1^H-NMR** (300 MHz, CD_2_Cl_2_) δ(ppm): 8.22 (d, *^4^J* = 1.5 Hz, 2H, Carb–H4), 8.13 (d, *^3^J* = 7.6 Hz, 2H, Carb–H5), 8.01 (d, *^3^J* = 3.9 Hz, 2H, Th–H3), 7.80 (s, 2H, BSeD–H5), 7.66 (dd, *^3^J* = 8.7 Hz, *^4^J* = 1.5 Hz, 2H, Carb–H2), 7.53–7.45 (m, 2H, Carb–H6), 7.44–7.38 (m, 4H, Carb–H1,8), 7.34 (d, *^3^J* = 16.1 Hz, 2H, Vinyl–H1), 7.30–7.20 (m, 4H, Vinyl–H2, Carb–H7), 7.16 (d, *^3^J* = 3.9 Hz, 2H, Th–H4), 4.17 (d, *^3^J* = 7.1 Hz, 4H, >N–CH_2_–), 2.17–1.97 (m, 2H, –CH<), 1.49–1.15 (m, 16H, –CH_2_–), 0.92 (*t*, *^3^J* = 7.4 Hz, 6H, –CH_3_), and 0.86 (*t*, *^3^J* = 7.1 Hz, 6H, –CH_3_). **^13^C-NMR** (75 MHz, CD_2_Cl_2_) δ(ppm): 158.54, 145.73, 141.82, 141.28, 138.18, 130.39, 128.62, 128.43, 127.37, 126.65, 126.26, 125.73, 124.71, 123.57, 123.11, 120.67, 119.53, 119.45, 118.83, 109.85, 109.75, 47.95, 39.85, 31.43, 29.23, 24.84, 23.47, 14.18, and 11.10. **HRMS** (ESI) calculated mass for C_58_H_58_N_4_S_2_Se^+^ [M]^+^: 954.3268 (100%), 955.3302 (63%), and 952.3276 (48%), found: 954.3224 (100%), 955.3277 (75%), and 952.3361 (60%).

**5,5’-bis((*E*)-2-(9-(2-ethylhexyl)-9*H*-carbazol-3-yl)vinyl)-2,2’-bithiophene (C5)**: The product was obtained in the homo-coupling side reaction of compound **7** when the preparation of compound **C2** was carried out in DMF using a procedure described by J. E. Baldwin et al. [41]. In this reaction, 0.65 g (0.96 mmol) of 9-(2-ethylhexyl)-3-(2-(5-(tributylstannyl)thiophen-2-yl)vinyl)-9*H*-carbazole (**7**) was used, and after purification, 150 mg of product **C5** was isolated in the form of an orange solid in 40% yield. Melting point: 168–170 °C. **^1^H-NMR** (300 MHz, CDCl_3_) δ(ppm): 8.18 (d, *^4^J* = 1.5 Hz, 2H, Carb–H4), 8.12 (d, *^3^J* = 7.7 Hz, 2H, Carb–H5), 7.61 (dd, *^3^J* = 8.7 Hz, *^4^J* = 1.5 Hz, 2H, Carb–H2), 7.53–7.42 (m, 2H, Carb–H6), 7.42–7.30 (m, 4H, Carb–H1,H8), 7.30–7.22 (m, 2H, Carb–H7), 7.23 (d, *^3^J* = 15.9 Hz, 2H, Vinyl–H1), 7.10 (d, *^3^J* = 15.9 Hz, 2H, Vinyl–H2), 7.10 (d, *^3^J* = 3.8 Hz, 2H, Th–H), 6.97 (d*^3^J* = 3.8 Hz, 2H, Th–H), 4.16 (d, *^3^J* = 7.1 Hz, 4H, >N–CH_2_–), 2.19–1.92 (m, 2H, –CH<), 1.50–1.12 (m, 16H, –CH_2_–), 0.92 (*t*, *^3^J* = 7.5 Hz, 6H, –CH_3_), 0.87 (*t*, *^3^J* = 7.1 Hz, 6H, –CH_3_). **^13^C-NMR** (75 MHz, CDCl_3_) δ(ppm): 142.78, 141.46, 140.89, 135.63, 129.67, 128.14, 126.37, 125.94, 124.37, 124.03, 123.34, 122.93, 120.50, 119.19, 119.17, 118.61, 109.38, 109.31, 47.67, 39.56, 31.15, 28.95, 24.54, 23.18, 14.17, 11.04. **HRMS** (ESI) calculated mass for C_52_H_56_N_2_S_2_^+^ [M]^+^: 772.3885, found: 772.3898; C_52_H_57_N_2_S_2_^+^ [M + H]^+^: 773.3958 (100.0%), 774.3991 (56.2%), and 775.4025 (15.5%), found: 773.3945 (100%), 774.3951 (50%), and 775.3914 (12%).

### 2.2. Electrochemistry

Cyclic voltammetry experiments were carried out on a classic three-electrode assembly with a Pt wire as the working electrode, a Pt spiral as the counter electrode, and Ag as the pseudoreference electrode on an Autolab PGSTATM101 potentiostat (Metrohm Autolab B.V., Utrecht, The Netherlands). The potential was calibrated versus a ferrocene/ferrocinium redox couple as an internal standard. Experiments were performed in dichloromethane (DCM) (Sigma-Aldrich, Chromasolv for HPLC) with 0.1 M tetrabutylammonium tetrafluoroborate (98%, Tokyo Chemical Industry Co., Nihonbashi-honcho, Japan) as the supporting electrolyte. All solutions were deaerated with Ar before measurements. Ionization potential (*IP*) was estimated from equation:*IP* = |e^−^|(5.1+*E_ox onset_*)(1)

Electron affinity *(EA)* was estimated from the equation:*EA* = |e^−^|(5.1+*E_red onset_*)(2)

Electrochemical band gap (*ΔEg_el_*) was estimated from the equation:*Δ**Eg*_*el*_ = *IP*−*EA*(3)

### 2.3. Photophysics

UV-Vis absorption spectra of dilute solutions (~10^−5^ M) and thin films were recorded on a Cary 5000 (Varian) spectrometer. Photoluminescence spectra of studied samples were performed at room temperature with an Edinburgh Instruments FLS980 fluorescence spectrometer (Edinburgh Instruments Ltd., Livingston, UK) with a Xe-lamp as an excitation source and an R-928 photomultiplier detector. The photoluminescence quantum yields (PLQY) of compounds were measured using an integrating sphere from Edinburgh Instruments with a BENFLEC interior coating. Fluorescence lifetimes of dilute solutions were acquired by time-correlated single photon counting (TCSPC) methods on an Edinburgh Instruments FLS980 fluorescence spectrometer.

### 2.4. Devices

OLEDs were fabricated by a hybrid spin-coating/evaporation method. The size of the pixels was 4.5 mm^2^. PEDOT:PSS (poly(3,4-ethylenedioxythiophene)–poly(styrenesulfonate), Heraeus Hanau, Germany) was used as a hole injection layer (HIL), and blends of PVK (Poly(9-vinylcarbazole), M_W_: 25,000–50,000) and PO–T2T (2,4,6-tris[3-(diphenylphosphinyl)phenyl]-1,3,5-triazine) doped with dyes were used as an emitting layer. Evaporated thin layers of PO–T2T were introduced as hole blocking layers (HBLs) and electron transport layers (ETLs). Cesium carbonate (Cs_2_CO_3_) and aluminum were used as the cathode. OLED devices were made on pre-cleaned and patterned indium-tin-oxide (ITO) substrates with a sheet resistance of 20 Ω/sq and an ITO thickness of 100 nm (Ossila Ltd, Sheffield, UK). PEDOT:PSS was spin-cast and annealed on a hotplate at 150 °C for 10 min to obtain 40 nm thin films. The active layers were spin-cast from chloroform solutions of PVK:PO–T2T (60:40 *w*/*w*) (12 mg/mL) with 8 wt% of dopant and then annealed at 60 °C for 20 min under a nitrogen atmosphere. All solutions were filtered before use with a PVDF or PTFE syringe filter with a 0.45 µm pore size. 50 nm layers of PO–T2T and cathode layers were thermally evaporated using a Prevac deposition system under a pressure of 10^−6^ mbar. An organic semiconductor (PO–T2T) and aluminum were deposited at a rate of 1 Ås^−1^, and the Cs_2_CO_3_ layer was deposited at a rate of 0.1–0.2 Ås^−1^. The thickness and roughness of the organic layers were controlled by a Bruker Dektak XT profilometer (Bruker Corporation, Billerica, Massachusetts, USA). The device characteristics were recorded using a Minolta CS-200 camera connected with a Keithley 2400 source measurement unit (Tektronix, Beaverton, Portland, OR, USA), whereas the electroluminescence spectra were recorded by Ocean Optic USB2000 and StellarNet SILVER-Nova Super Range TEC spectrometers. All materials were purchased from Sigma Aldrich (Saint Louis, MO, USA) or Lumtec (Luminescence Technology Corp., New Taipei City, Taiwan).

## 3. Results

### 3.1. Synthesis

The emissive materials were obtained by Stillecoupling between different acceptor units and stannylated D–π arm-containing thiophene vinyl and carbazole units. D–π arms were obtained via the multistep synthetic route shown in Figure 2.

The first step of the synthesis was performed where the nitrogen atom of 9*H*-carbazole (**1**) was alkylated by a nucleophilic substitution reaction (S_N_2). Next, the obtained alkylcarbazol (**2**) was formylated using the Vilsmeier–Haack reaction, and the resulting aldehyde (**3**) was reduced using sodium borohydride to yield the alcohol (**4**). This alcohol was converted to a phosphonium salt (**5**) using triphenylphosphine hydrobromide, and the resulting product (**5**) was condensed with 5-bromo-2-formylthiophene through a Wittig reaction to obtain a vinyl bond between the carbazole and thiophene units. 3-(2-(5-bromothiophen-2-yl)vinyl)-9-(2-ethylhexyl)-9*H*-carbazole (**6**) was formed only when a weak base was used (K_2_CO_3_), and all attempts using stronger bases, such as MeONa or *t*-BuOK were unsuccessful. The Wittig reaction gave compound **6** as a mixture of two isomers (*E* and *Z*) which were not separated into individual components, and were used in such a form in the next stages. Both isomers were converted into their stannyl derivatives (**7**) by two subsequent reactions: Halogen-lithium and lithium-tin exchange. The final stage involved Stille cross-coupling and isomerization of vinyl bonds to give the most thermodynamically-stable isomer (*E*,*E*) of the final products (**C1**, **C2**, **C3,** and **C4**). It is also worth noting that the last step must be performed in a nonpolar solvent (e.g., toluene) because when the reaction was carried out in polar solvents such as DMF, compound **C5** was the main product, which is a product of the homo-coupling of compound **7**. All the final compounds except **C3** show good solubility in common organic solvents, such as DCM, toluene, and chloroform.

### 3.2. Electrochemical Properties

All studied compounds contained two identical electron-donor moieties and different electron-acceptor units. Electrochemical measurements were used to estimate the influence of these units using cyclic voltammetry (CV) in a DCM/Bu_4_NBF_4_ electrolyte, and the results are shown in Figure 3 and summarized in Table 1. All samples showed quasi-reversible reduction peaks. The reduction onset potential (*E*_red onset_) varied from −1.42 V to −1.76 V, depending on the central electron-acceptor unit. The reduction of the reference compound C5 was recorded at −2.25 V. The results are in good agreement with other donor–acceptor compounds in which the type of electron-deficient unit determined the reduction behavior and the energy of the LUMO (*E*_LUMO_) [7,42,43]. The reduction was related to the insertion of an electron into the LUMO [44,45].

Analyzing the CV curves in the oxidation range revealed large differences between the compounds. The oxidation onset potential (*E*_ox onset_) ranged from −0.12 V for **C4** up to 0.4 V for **C1**. The large difference of *E*_ox onset_ between these two similar compounds exceeded 0.5 V, while both compounds exhibited nearly the same *E*_red onset_. The large differences in *E*_ox onset_ in all studied compounds clearly indicates that electron-acceptor units have a significant impact on the energy of the HOMO (*E*_HOMO_), which determines properties such as the *IP* and oxidation behavior.

*E*_red onset_ and *E*_ox onset_ were used to estimate the electrochemical *EA* and *IP* (Figure 4) [46]. The values of electrochemically-estimated *EA*, *IP*, *E*_HOMO_, and *E*_LUMO_ are summarized in Table 1. The acceptor units affected both *IP* and *EA*, and the analysis of the electrochemical results revealed that the acceptor unit significantly altered *IP* and *E*_HOMO_. The direction of the changes was different for different acceptors. In compounds **C1**, **C2**, and **C3**, the HOMO was stable compared with **C5**, while **C4** showed a much higher *E*_HOMO_ value. **C4**, which contained a benzselenadiazole unit, exhibited an *IP* shift of 4.98 eV. These results clearly indicate the unusual properties of benzselenadiazole, which simultaneously increased *E*_HOMO_ and lowered *E*_LUMO_, leading to low molecular weight material with a narrow energy gap (Δ*Eg*_el_) of 1.47 eV. Compared with other donor–acceptor π-conjugated molecules, such a narrow Δ*Eg*_el_ is usually only seen in highly-conjugated polymers [47,48,49,50]. The results reveal the potentially beneficial properties of benzselenadiazole-based conjugated materials, and these results are in good agreement with other research [51,52,53,54,55,56].

### 3.3. Photophysical Properties

The normalized optical absorption spectra of toluene solutions and spin-coated thin films of the studied materials are shown in Figure 5. Molecules showed two main absorption regions: The main lower-energy bands from 451–583 nm and higher-energy bands from 200–300 nm and 350–550 nm (Table 2). The absorption spectra of the studied materials were clearly correlated with their donor–acceptor–donor (D–A–D) structures, where 9-(2-ethylhexyl)-9H-carbazole was introduced as a donor unit linked with different acceptors by a π-linker. All molecules exhibited high-energy absorption bands centered in the 200–340 nm region (Figure 5b), which was correlated with the absorption of the carbazole moiety [57,58]. The **C5** and **C1** spectra look similar, with one main band located at 451 nm in **C5** which is red-shifted in the **C1** spectrum. This absorption band was attributed mainly to the bithiophene core of the molecules [59]. Molecules **C2**, **C3**, and **C4** showed dual-band absorption profiles with their lowest-energy absorption peaks positioned at 520 nm, 539 nm, and 583 nm, respectively. These peaks originated mainly from an extensive π-conjugated system involving the acceptor moiety quinoxaline core [60], difluoro-2,1,3-benzothiadiazole [61,62], and 2,1,3-benzoselenadiazole [63,64]. The red shift observed in the lowest-energy absorption band of **C4** correlated closely with the electrochemical results, and is explained by a decrease in the degree of aromaticity in the acceptor unit when the sulphur was replaced with selenium [65]. This destabilized the HOMO and simultaneously stabilized the LUMO, which led to a red-shift in the absorption spectrum.

It has been suggested in works devoted to the photophysics of D–A type molecules that the low-energy band of the absorption spectrum arises due to intramolecular charge transfer [66,67]. Therefore, the absorption and emission spectra of studied compounds were obtained in solvents with different polarities, and the absorption spectra remained unchanged (Appendix A), which may suggest that polarity doesn’t affect the energy of ground states.

The solid-state absorption spectra displayed a red shift compared with spectra in dilute solutions due to interactions between molecules occurring in neat films. From the long wavelength absorption edge of the UV−VIS spectra in the solid state, the optical band gaps of emitters were estimated. The *E*_g_ values are given in Table 1, where it was noticed that different molecular cores influenced optical properties of studied D–A–D compounds, such as optical band gaps and as a consequence the color of emitted light can be tuned (Figure 6).

The photoluminescence (PL) spectra of compounds in toluene solution and a zeonex matrix are shown in Figure 6, and the photoluminescence maxima and photoluminescence quantum yields (PLQY) in different solvents are collated in Table 2. The designed molecules showed fluorescence with a large Stokes shift, which covered a broad part of the spectrum from the green to near-infrared regions. A vibronic structure of the fluorescence spectra of **C5** and **C1** in toluene was observed, with emission bands centered at 505 nm, 541 nm, and 580 nm for **C5** and emission maxima at 568 nm, 612 nm, and 665 nm for **C1**. In contrast, molecules **C2**, **C3**, and **C4** displayed one broad main band with emission maxima at 651 nm, 660 nm, and 740 nm, respectively (Figure 6a, Table 2). These results, together with the Stokes shift, much smaller in the case of **C1** and **C5** molecules, may suggest more rigid structure, and smaller geometry and electronic density distribution changes during the excitation in the two molecules. However, in a non-polar rigid medium (zeonex), these emitters demonstrated blue-shifted emissionin comparison to the emission observed in solvent. This rigidochromic effect suggests the intramolecular charge transfer (CT) character of the emissive state, which seems to be confirmed by the solvatochromism experiment. All molecules except **C5** displayed a positive solvatochromic effect where the emission spectra shifted to a longer wavelength as the solvent polarity increased (Table 2, Appendix A). Meanwhile, the PL spectra of the neat films (Appendix A) exhibited a strong red shift and a high amount of excited state quenching, due to strong intermolecular interactions. The PL spectra of **C5** did not change significantly when the molecule was placed in different environments. This may suggest that excitation did not change the dipole moment in molecules without an acceptor unit, and a CT state was not created. Moreover, in the case of **C5**, the highest photoluminescence quantum yield was determined to be practically independent of solvent nature (Table 3).

Fluorescence decay measurements (Appendix A) were performed for dilute solutions at room temperature. The decay profiles were observed to occur on the nanosecond scale, which was assigned to fluorescence emission. The summarized lifetime (τ) values of the molecules in dilute toluene, chloroform, DCM, and THF are given in Table 3. It should be noticed that for compounds **C1**, **C2**, **C3**, and **C4**, it was difficult to fit the emission decay with a monoexponential function, where a very small contribution of the second time was detected. The origin of it is unclear. However, in the case of the compound **C5**, emission decayed monoexponentially (for details, see Appendix A). It can be seen that the radiative rate constant, obtained from the PLQY/τ ratio, systematically decreased upon passing from **C5**, **C1**, **C3**, **C2**, to **C4**. Evidently, the introduction of a selenium atom in the acceptor unit decreased the radiative rate constant. Such behavior may be interpreted in terms of relative twisting of the donor and acceptor moieties in the excited state. It is generally known that the radiative rate constant is a function of the twist angle (Θ) and decreases when the value of Θ changes from 0° to 90°, where the two moieties involved in the charge transfer are orbitally decoupled and, as a consequence, the transition is forbidden and *k*_r_ becomes very small. However, additional experiments and/or calculations are needed to confirm this hypothesis. On the other hand, the very low emission quantum yield observed for **C4** could be related to activation of some radiation-less processes. One of them could be an intersystem crossing phenomenon induced by a spin-orbit coupling enhancement, produced by a heavy atom. Additionally, it is known that in long-wavelength emitters, enhanced internal conversion is responsible for quenching emissive states [68].

To reduce the effects of concentration quenching and to evaluate the potential use of low concentrations of doping complexes for solution-processed OLEDs, exciplex hosts were used. Exciplex forming blend of donor and acceptor materials have been used as efficient hosts for fluorescent emitters. Such systems provide efficient charge carrier balance, which results in a long and stable operating lifetime of OLEDs [69]. Recently, Pander et al. reported a novel TADF exciplex comprised of a PVK polymer (donor) and a PO–T2T acceptor, and its further application in solution-based OLEDs [70]. The corresponding exciplex was used as a host for synthesized molecules. Moreover, the emission of a green PVK:PO–T2T exciplex overlapped with the absorption spectra of synthesized complexes, which is necessary for efficient Förster resonance energy transfer (FRET) (Appendix A) [71]. Much better matching of these spectra was observed in the case **C2**, **C3**, and **C4**, and the largest overlap was achieved by **C4**. The photoluminescence spectra of thin films of PVK:PO–T2T doped with 1, 2, 5, 8, and 10 wt% of studied compounds are shown in Appendix A and Figure 7a. The studied materials exhibited similar behavior, and at a low doping concentrations, the PL spectra contained two emission peaks responsible for PVK:PO–T2T exciplex emission (530 nm) and dopant fluorescence (Appendix A). Complete energy transfer was achieved at 8 and 10 wt% emitter doping concentrations in PVK:PO–T2T. Interestingly, as the dopant concentration increased in the exciplex, a bathochromic shift was observed in the PL spectra (Appendix A). Additionally, the photoluminescence quantum yields of all thin films were similar and were in the range 4–5%, which suggests strong concentration-induced quenching in the solid state.

### 3.4. Device Fabrication

To unlock the potential of study’s materials, they have been implemented in OLEDs. Since **C1–C5** were designed and synthesized for solution-processable techniques, they were incorporated as emitting spices in the polymer light-emitting diodes. The structure of the OLED prototypes were as follows: ITO/PEDOT:PSS (40 nm)/8 wt% (**C1**–**C5**) in PVK:PO–T2T (80 nm)/PO–T2T (50 nm)/Cs_2_CO_3_ (2 nm)/Al (100 mm). The first layer of PEDOT:PSS (poly(3,4-ethylenedioxythiophene)-poly(styrenesulfonate)) was used as a hole injection layer and was spin cast on ITO substrates from an aqueous solution. Next, the active layer consisted of a blend of low–molecular-weight PVK (Poly(9-vinylcarbazole)) and PO–T2T (2,4,6-tris[3-(diphenylphosphinyl)phenyl]-1,3,5-triazine) doped with 8 wt% of the respective molecules was spin coated on the PEDOT:PSS layer from chloroform solution. A ratio of 60:40 of PVK:PO–T2T was chosen to ensure balanced charge carrier transport to the emitters. The doping concentrations of **C1**–**C5** were chosen due on the base of phophysical studies to guarantee complete energy transfer from the exciplex (PVK: PO–T2T) host to the emitters. Active three-component layers were spin coated by a chloroform solution on the PEDOT:PSS layer. Such polymer light-emitting layers have a good film-forming property, good carrier transporting characteristics, and can be easily processed by different solution-processable techniques (e.g., inkjet printing and slot die coating). Furthermore, a thermally-evaporated 50 nm layer of PO–T2T was introduced as an electron transport layer.

The OLED characteristics are presented in Figure 8, and the energy levels of organic semiconductors used in the devices are inserted in Figure 8a. The normalized electroluminescence (EL) spectra of OLEDs (Figure 7b) showed main emissive bands centered at 565 nm, 602 nm, 680 nm, 680 nm, and 765 nm for devices based on **C5**, **C1**, **C2**, **C3**, and **C4**, respectively. EL was observed in a large part of the spectrum, from green to near-infrared emission, and did not contain emission from additional layers or from the host. This indicated that excitons were fully transferred from the host to the emissive species, or the charge carriers’ trapping process plays important role. Nevertheless, fabricated devices exhibited high current density at high voltage operation regimes (Figure 8a). It should be explained that OLEDs prototypes contained only three organic layers; the introduction of additional layers, such as an electron-blocking one, can improve charge carrier balance and current density characteristics.

OLEDs made with **C5** emitter showed a rather low turn-on voltage (*V*_on_) of 5 V, compared with OLEDs made from **C1**–**C3** emitters, which had turn-on voltages of 7 V, and for near-infrared **C4** molecule, −15 V (Figure 8b). The best performance in this work displayed the green solution processed OLEDs with the **C5** molecule; the maximum brightness (L_max_) and maximum current efficiency (_ηL,max_) were 1452 cd m^−2^ at 14 V and 0.44 cd A^−1^, respectively. The device fabricated with **C1** emitter also exhibited a comparable performance with η_L,max_ of 0.44 cd A^−1^ and L_max_ of 726 cd m^−2^ at 16 V. The group of red polymer OLEDs made with **C2** and **C3** molecules showed similar electroluminescence spectra with maximum emission at 680 nm but with worse work parameters. Maximum current efficiency (η_L,max_) and maximum brightness (L_max_) were found to be 0.19 cd A^−1^ and 255 cd m^−2^ at 17 V for the **C2** device; η_L,max_ and L_max_ were −0.055 cd A^−1^ and 146 cd m^−2^ at 16 V for device **C3**. The implementation of **C4** molecules as the emitter in polymer OLEDs allowed us to achieve near-infrared EL emission with a maximum at 765 nm. However, poor performance parameters, such as an L_max_ of 20 cd m^−2^ at 22V and an η_L,max_ less than 0.002 cd A^−1^ were obtained. Such extremely low efficiency can be partially explained that emission of the device is on the edge of visible spectra and sensitivity of photometric equipment used in this work.

To sum up, a series of solution-processed, three layer OLEDs have been fabricated, where green polymer exciplex PVK:PO–T2T was incorporated as a host, and studied materials as the emitters. It was shown that, in the series of donor–acceptor D–π–A–π–D emitters, different acceptors had influence not only on photoluminescence but also on electroluminescence properties. This allowed the shift of EL spectra of OLEDs from green to near-infra red color. Studied OLEDs displayed extremely low current efficiency; for that reason external quantum efficiency and power efficiency of devices were not discussed. The low photoluminescence quantum yield, strong concentration quenching in the solid state and fluorescence nature of emitters may be responsible for the limited the efficiency of fabricated polymer OLEDs.

## 4. Conclusions

The series of new D–π–A–π–D compounds were synthesized, and their optical, photophysical, and electrochemical properties were studied. It was found that introducing different acceptor moieties in studied D–π–A–π–D compounds allows a narrow optical bandgap, and the consequential shift fluorescence emission from the green to near-infrared part of spectra. Moreover, D–π–A–π–D materials displayed a positive solvatochromic effect, which suggests charge transfer character of the emissive states. Besides that, electrochemical measurements revealed that the acceptor units significantly changed not only the reduction potential related to the LUMO, but also the oxidation potential related to the HOMO. The oxidation onset depended on choice of acceptor, and ranged from −0.12 V for **C4** up to 0.4 V for **C1**. 2,1,3-Benzoselenadiazole was identified as an acceptor unit which modified both the LUMO and HOMO energy levels and led to compounds with narrow energy gaps. Finally, the studied materials were implemented in prototype OLEDs as the emitters, together with green polymer exciplex PVK:PO–T2T as a host. The maximum EL intensities of solution-processable polymer devices based on **C5**, **C1**, **C2**, **C3**, and **C4** emitters were observed at 565 nm, 602 nm, 680 nm, 680 nm, and 765 nm, respectively.

## Figures and Tables

**Figure 1 nanomaterials-09-01179-f001:**
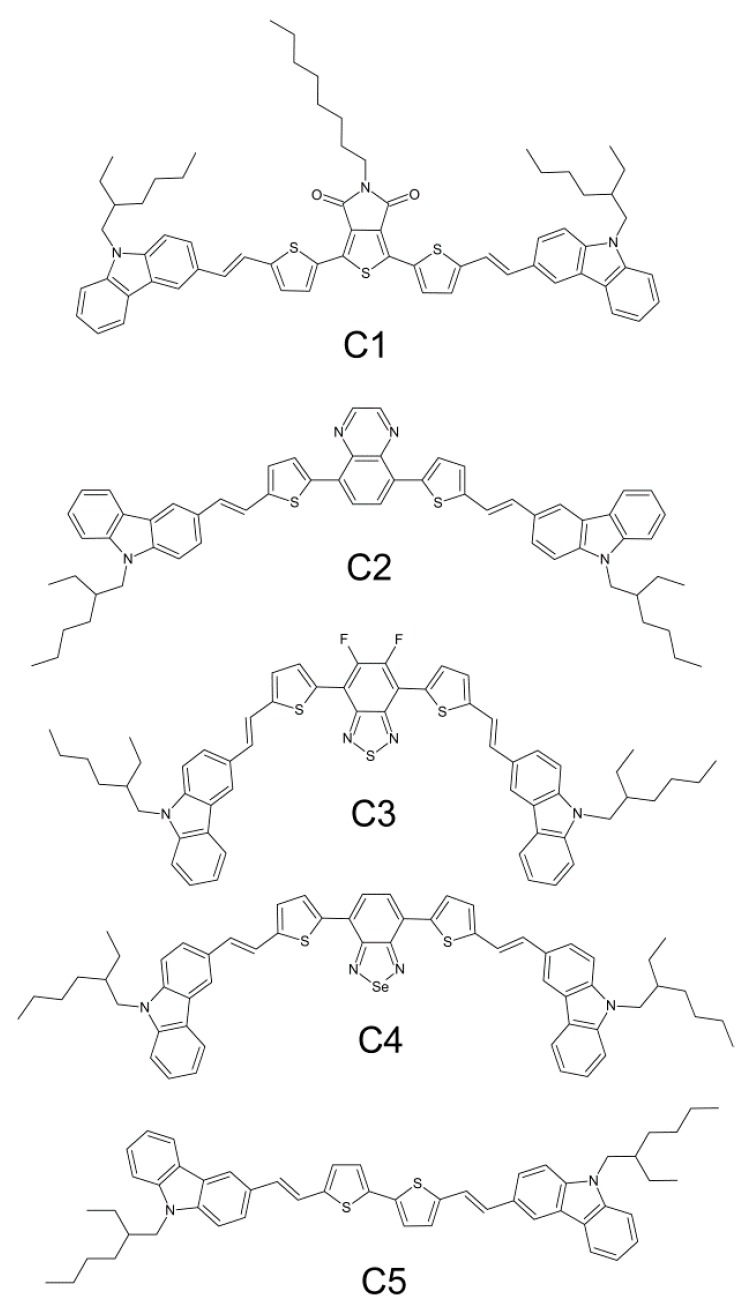
Chemical structures of compounds **C1**, **C2**, **C3**, **C4** and **C5**.

**Figure 2 nanomaterials-09-01179-f002:**
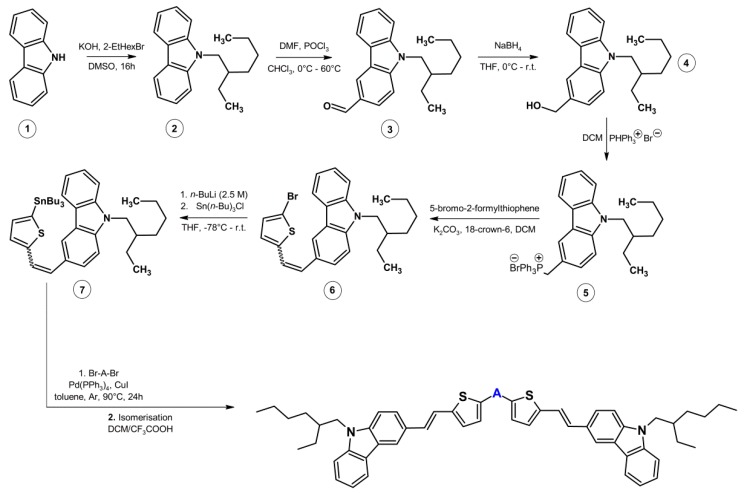
Synthesis procedure.

**Figure 3 nanomaterials-09-01179-f003:**
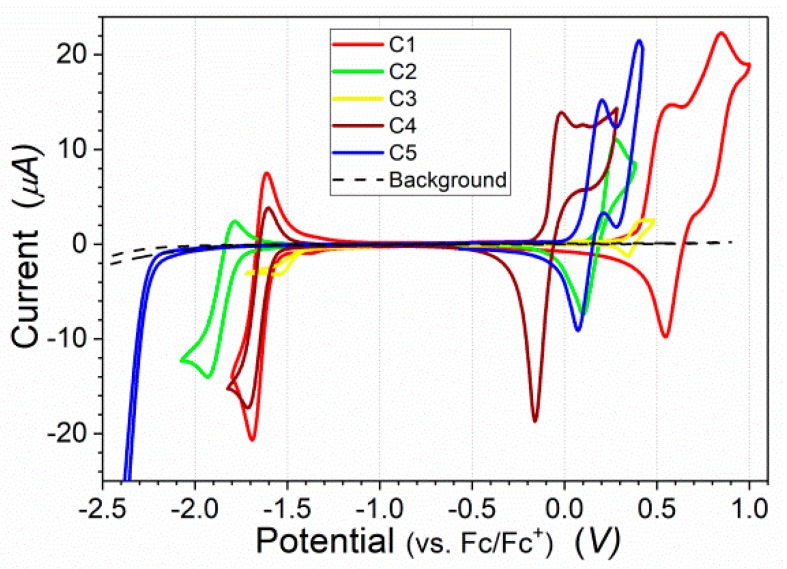
Cyclic voltammetry of 4 mM solutions of **C1**, **C2**, **C4**, **C5** or a saturated solution of **C3** in DCM/Bu_4_NBF_4_ electrolyte, at a potential sweep rate of 50 mV/s.

**Figure 4 nanomaterials-09-01179-f004:**
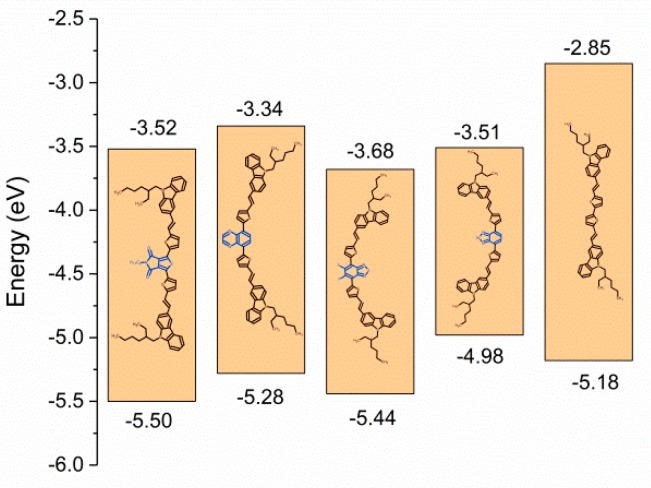
Energy diagram of compounds **C1**–**C5**.

**Figure 5 nanomaterials-09-01179-f005:**
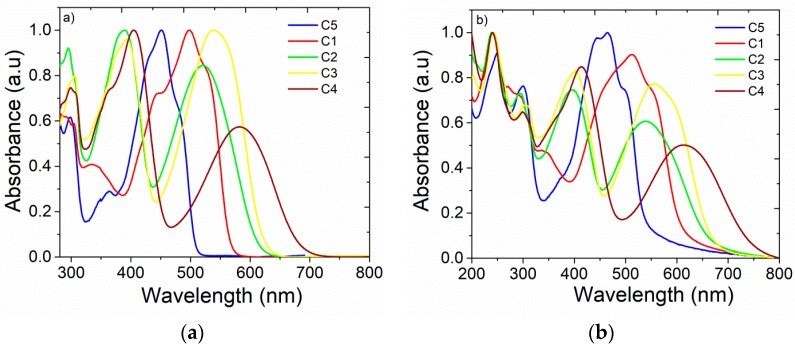
Normalized absorption spectra of dilute solutions (10^−5^ M in toluene) (**a**), and spin-coated thin films (**b**) of the compounds.

**Figure 6 nanomaterials-09-01179-f006:**
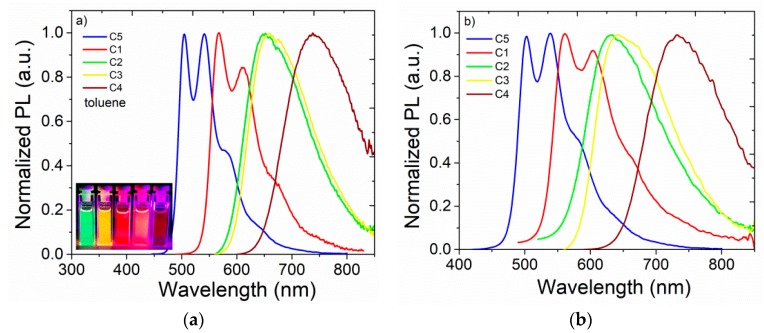
Normalized photoluminescence (PL) spectra of the dilute solutions (10^−5^ M in toluene) (**a**), and 1 wt% of emitters in a zeonex matrix (spin-coated thin films) (**b**).

**Figure 7 nanomaterials-09-01179-f007:**
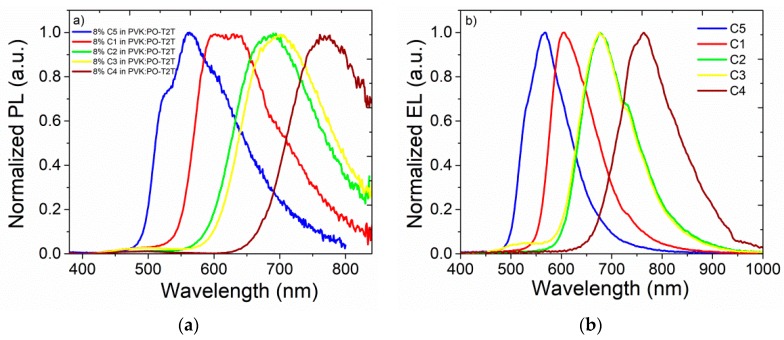
Photoluminescence spectra of 8 wt% **C1–C5** doped in PVK:PO–T2T host (**a**), and the electroluminescence spectra of fabricated OLEDs (**b**).

**Figure 8 nanomaterials-09-01179-f008:**
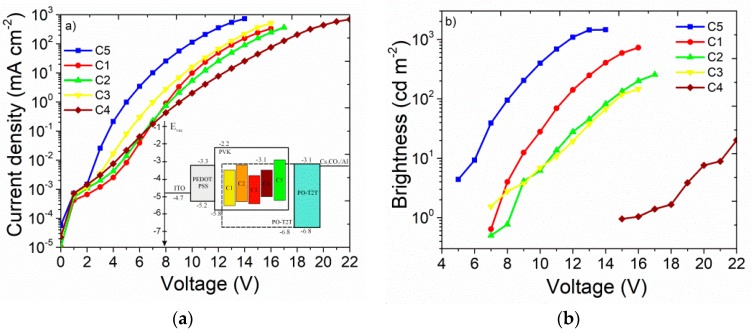
Current density−voltage (insert—energy band diagram of OLEDs) (**a**), and luminance−voltage characteristics (**b**) of fabricated devices.

**Table 1 nanomaterials-09-01179-t001:** Electrochemical and optical properties.

	^1^ *E_ox onset_*	^2^ *E_red onset_*	^3^ *IP*	^4^ *EA*	^5^ *E_HOMO_*	^5^ *E_LUMO_*	^6^ *ΔEg_el_*	^7^ *ΔEg_opt_*
Compound	[V]	[V]	[eV]	[eV]	[eV]	[eV]	[eV]	[eV]
**C1**	0.4	−1.58	5.50	3.52	−5.50	−3.52	1.98	2.03
**C2**	0.18	−1.76	5.28	3.34	−5.28	−3.34	1.94	1.86
**C3**	0.34	−1.42	5.44	3.68	−5.44	−3.68	1.76	1.85
**C4**	−0.12	−1.59	4.98	3.51	−4.98	−3.51	1.47	1.65
**C5**	0.08	−2.25	5.18	2.85	−5.18	−2.85	2.33	2.27

^1^*E*_ox onset_ is the onset oxidation potential; ^2^
*E*_red onset_ is the onset reduction potential; ^3^
*IP* is the ionization potential estimated from the equation *IP* = |e^−^|(5.1 + *E*_ox onset_); ^4^
*EA* is the electron affinity estimated from the equation *EA* = |e^−^|(5.1 + *E*_red onset_); ^5^
*E*_HOMO_ and *E*_LUMO_ were electrochemically-estimated approximate values of the HOMO and LUMO energy; ^6^ Δ*E*g_el_ is the electrochemical band gap, calculated from *ΔEg_el_* = *IP* – *EA*; and ^7^
*E*g_opt_ is the optical band gap, as estimated from the edges of electronic absorption spectra of thin films.

**Table 2 nanomaterials-09-01179-t002:** The photophysical properties of **C1**–**C5** in dilute solution (10^−5^ M in toluene, chloroform, THF, and DCM) and in the solid state.

	λ_abs_,_tol_ [nm]	λ_abs_,_film_ [nm]	λ_PLzeon._ [nm]	λ_PL,tol_ [nm]	λ_PL,chlor_ [nm]	λ_PL,THF_ [nm]	λ_PL,DCM_ [nm]	λ_PLfilm_ [nm]
**C5**	451	465	502	505	509	506	510	528
**C1**	498	514	560	568	582	585	592	685
**C2**	520	540	633	651	670	675	679	725
**C3**	539	558	640	660	695	701	707	740
**C4**	583	614	733	740	770	776	782	835

**Table 3 nanomaterials-09-01179-t003:** Photoluminescence quantum yield (Φ_PL_), lifetime (τ ns) and radiative, non-radiative constants of fluorescence for C1–C5 in diluted solution.

Mol	Solvent	Φ_PL_, %	τ ns	Radiative Rate Constant, (10 ^9^ s^−1^)	Non-Radiative Rate Constant, (10 ^9^ s^−1^)
**C5**	TOL	52	0.9	0.57	0.54
	CHL	42	0.8	0.52	0.75
	DCM	50	0.9	0.56	0.56
	THF	52	0.9	0.58	0.53
**C1**	TOL	48	1.3	0.37	0.40
	CHL	48	1.2	0.40	0.43
	DCM	42	1.1	0.38	0.52
	THF	41	1.1	0.37	0.54
**C2**	TOL	34	1.9	0.17	0.35
	CHL	33	2.1	0.16	0.32
	DCM	31	1.9	0.16	0.36
	THF	32	2.1	0.15	0.32
**C3**	TOL	45	2.3	0.20	0.24
	CHL	45	2.9	0.16	0.16
	DCM	38	2.5	0.15	0.25
	THF	39	2.7	0.14	0.23
**C4**	TOL	17	2.2	0.07	0.34
	CHL	7	1.1	0.06	0.75
	DCM	4	1.0	0.04	0.95
	THF	7	1.5	0.047	0.62

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
