# Peer review of "The Effect of Acceptor Structure on Emission Color Tuning in Organic Semiconductors with D–π–A–π–D Structures"

_nanomaterials, 2019, doi:10.3390/nano9081179_

Round 1

Reviewer 1 Report

This manuscript authored by Prof. Ledwon et al. describes a series of D-π-A-π-D compounds, exhibitingfluorescence and electroluminescence from green emission to near-infrared region. All syntheses and characterizations were properly done and straightforward. The work was clearly presented and detailed descriptions were fully supported by various experiments and device fabrication. After minor revision according to the comments below, this manuscript should be acceptable in the Nanomaterials.
1) HR-MS and Elementary data of C1 are missing in the manuscript and supplementary file.
2) 13C NMR data of C3  should be presented.

Reviewer 2 Report

Decision: Major revision

Comments:

1.       Authors should discuss more about OLED dopants while explaining its working principle (Fluorescent, phosphorescent and TADF dopants) in introduction section.

2.       Some references for TADF emitters with donor-acceptor (D-A and D-A-D) molecular design:

l  10.1039/C8TC06293E

l  10.1039/C7TC02156A

l  10.1039/C9TC02491C

l  10.1021/acsami.9b06364

3.       Introduction must include a detail discussion about Donor-acceptor type molecules in OLED.

4.       Authors should provide elemental analysis data of all final molecules and analyzed data should be added in supporting information.

5.       DFT calculation should be made and discuss more while comparing with experimental values.

6.       Spatial distribution of frontier molecular orbitals can be added to understand the donor- acceptor moieties.

7.       PL decay curves are presented in supporting information, but there is not much discussion found in manuscript.

8.       Are these compounds exhibit TADF properties? explain more about it

9.       There is no device efficiency data found, so authors must provide power and external quantum efficiencies of each devices?

10.   The current efficiencies of devices were noticed too low with high voltage, authors should explain about it?

11.   Why non-radiative decay rate is higher than that of radiative decay rate?

12.   Why C4 emitter showed low photoluminescent quantum yield and high non radiative decay rate? Authors should give more details on this

13.   Did they show any quenching effect with different polarity of solvents?

14.   (Figure 6) PL spectra od C1 and C5 show side emission peaks, what is the reason behind that?

15.   Authors must add more discussion about D-π-A-π-D types molecules and compare the device performances with current study.

Discussion of device efficiencies while comparing with other reported molecules are very import in case of OLED research. Because readers will get clear idea about molecular design of highly efficient materials when authors provide comparison with similar reported molecules.

Author Response

Authors should discuss more about OLED dopants while explaining its working principle (Fluorescent, phosphorescent and TADF dopants) in introduction section.

Following the recommendation of the referee introduction section was modified and additional discussion was added. Please see improved manuscript.

"Emitters for OLEDs can be divided into inorganic semiconductors, metal complexes, and organic compounds,[10,11] and there is growing interest in completely organic light-emitting diodes (OLEDs). Currently, most commercially-available OLEDs are mainly based on Ir or Pt emitters because of their high efficiency and working stability.[12–14] In such devices the emission layer consists of the host organic material and guest phosphorescence dopant. The main drawback of such devices is the use of precious metals which results in irreversible damage to the environment. One of the most promising solutions is the use of fully organic fluorescent emitters which exhibit thermally activated delayed fluorescence (TADF).[15–17] In the typical fluorescence-based devices a neat film of the emitter or the host-guest system can be used as an emission layer.[18] In addition, a common fluorescent molecule combined with a TADF emitter can be used in hyperfluorescent OLEDs.[19] In such devices, a TADF emitter acts as a sensitizer that harvests excitons and transfers them to a fluorescent emitter by Förster energy transfer.[20] This can be an important strategy that will enable the fabrication of efficient and stable fully-organic OLEDs"

Some references for TADF emitters with donor-acceptor (D-A and D-A-D) molecular design:  10.1039/C8TC06293E; 10.1039/C7TC02156A; 10.1039/C9TC02491C;10.1021/acsami.9b06364

We are grateful to the reviewer for these new and interesting articles about TADF. Please notice, that TADF phenomena are not a scope of our article and it was not subject of our research. Also, thermally activated delayed fluorescence was not mentioned in keywords. A short discussion about TADF was introduced in the introduction as an example of another route to achieve NIR emitter. Moreover, conventional TADF can serve as a host for pure fluorescence emitters and can be realized in efficient hyperfluorescent OLEDs. Such a brief description can give to reader’s information about the current stage of developing NIR emitters. We have cited a 3 articles concerning this topic, [15] for the first time demonstrated the implementation of TADF in OLEDs, [16] it is a comprehensive review about TADF emitters and [17] showing recent progress in NIR TADF emitters. We believe that all these references are sufficient for such short description and additional literature will over saturate literature.  

Introduction must include a detail discussion about Donor-acceptor type molecules in OLED.

Following this recommendation of the reviewer we modified the abstract, introduction and added a discussion about donor-acceptor type molecules. The added text is as follows: 

"One of the most promising strategies used to obtain organic molecules with emission in the red and infrared range is the connection of a strong electron-acceptor (A) unit with a strong electron-donor (D) unit to form polymer or oligomer with structure D-A, D–A–D or A–D–A.[26] The structure can also be modified by adding a π-conjugated bridge (π) which can increase the effective conjugation length. The choice of the appropriate D, A and π units is crucial to control over the material functionality. Carbazole is one of the most frequently used donor units in many D-A type of dyes owning to its advantages such as high luminescent efficiency and broad ability to modify its structure.[27] So far different electron-accepting moiety and their derivatives including benzoselenadiazole,[28] benzothiadiazole,[29] quinoxaline,[30] thienopyrroledione[31] were reported as acceptor units with low LUMO level. However, there is only limited number of reports that analyze the impact of different types of strong acceptor units on luminescence properties.[32]" 

Authors should provide elemental analysis data of all final molecules and analysed data should be added in supporting information.

The compounds were characterized by 1H NMR, 13C NMR and HRMS. We believe that all these experimental techniques are ample to confirm the structure of new compounds and their purity. Elemental analysis is important technique and mostly used to determine the purity. However, due to the lack of quick access to this type of equipment, we are not able to obtain the results of elemental analysis during 5 days required for the revision of the manuscript so we believe that HRMS can be used alternatively. Some kind of suggestion of the purity of synthetized compounds could be also the results of melting point experiment.

DFT calculation should be made and discuss more while comparing with experimental values.

We agree with referee comment, and we have no doubt that DFT calculation would be a good complement to the experiment and would be very useful and helpful in interpretation of obtained results. Unfortunately, during 5 days required for the revision of the manuscript, we are not capable to perform proper DFT calculations of all 5 molecules. Competent calculations, analysis and providing the final results require more time and involvement of additional Researchers.

Spatial distribution of frontier molecular orbitals can be added to understand the donor- acceptor moieties.

We agree that spectral properties are strongly dependent on frontier molecular orbitals’ (FMOs) properties. Therefore, DFT calculations could be employed to investigate HOMO and LUMO of the studied molecules and the electronic density redistribution during the excitation would be informative in studying donor-acceptor properties. However, as it was mentioned in answer to the previous question, we are not able to make DFT calculation during 5 days required for the revision of the manuscript. We believe that, our experimental results, among them positive solvatochromic effect (Figure S17), confirm the charge transfer (CT) character of materials resulting from their donor-acceptor structure

PL decay curves are presented in supporting information, but there is not much discussion found in manuscript.

In order to answer to this question and following the recommendation of the referee we have modified slightly discussion concerning fluorescence decays.

Are these com pounds exhibit TADF properties? explain more about it

The presented materials do not exhibit TADF behaviour; consequently, TADF was not a scope of our article. In our article, the photoluminescence time decays were performed to investigate the nature of the emission and no evidence of delay fluorescence was found. Short lifetimes (in nanosecond range) suggest fluorescencent nature of studied molecules.

There is no device efficiency data found, so authors must provide power and external quantum efficiencies of each devices?

In part of the article concerning device fabrication and characterization, it was discussed only turn-on voltage, the maximum brightness and maximum current efficiency for OLEDs. Maximum current efficiency for the best devices based on C1 and C5 emitters were found at 0.44 cd/A, at the same time OLEDs made with C2, C3, C4 emitters showed even worse performance. Typically, external quantum efficiency (EQE) and power efficiency (PW) are calculated from the current efficiency. It follows that EQE and PE will exhibit extremely low values ( EQE below 0.1 %). These results were not included, because it is difficult to analyse and compare them. The performance of electroluminescence devices based on C1-C5 emitters is far away from the state-of-the-art OLEDs. The message behind this is that change of acceptor in D-A-D molecules has an influence not only on photoluminescent but also electroluminescencent properties. Some explanation has been added in manuscript text.

The current efficiencies of devices were noticed too low with high voltage, authors should explain about it?

We are not sure if we properly understand the referee’s question.

In OLEDs part, it was discussed only maximum values of current efficiency and maximum values of brightness at some given voltages. It is evident that maximum brightness for electroluminescent devices are obtained at high voltages (when brightness increased energy consumption also increased). At the same time, a maximum of the current efficiency of devices are not obtained at high voltage where consumption of energy is big enough. Maximum current efficiency can be found at the point where conversion of the number of injected electrons to photons is most efficient. Maybe this question arises from the unproper construction of the sentences, so to make it more clear OLEDs discussion part was modified.

Why non-radiative decay rate is higher than that of radiative decay rate?

Our photophysical studies revealed that for investigated molecules, with the exception of C4 , the probablility of radiative and nonradiative deactvation processes is comparable. For some reason the radiative rate constant in C4 is much smaller than for the rest of tested molecules. We have provided some tentative explanation of that in manuscript.

“In some cases the nonradiative decay rate is higher than radiative one and it means that some nonemisive pathways can play important role or even dominate in the deexcitation process. “

Why C4 emitter showed low photoluminescent quantum yield and high non radiative decay rate? Authors should give more details on this

Following the recommendation of the referee we have added a tentative explanation about the low quantum yield of C4. The added text is as follows:

“ It can be seen that the radiative rate constant, obtained from the PLQY/ τ ratio, systematically decreased upon passing from C5, C1, C3, C2, to C4. Evidently, the introduction of a selenium atom in the acceptor unit decreased the radiative rate constant. Such behavior may be interpreted in terms of relative twisting of the donor and acceptor moieties in the excited state. It is generally known that the radiative rate constant is a function of the twist angle (Θ) and decreases when the value of Θ changes from 0° to 90°, where the two moieties involved in the charge transfer are orbitally decoupled and, as a consequence, the transition is forbidden and kr becomes very small. However, additional experiments and/or calculations are needed to confirm this hypothesis. On the other hand, the very low emission quantum yield observed for C4 can be related to activation of some radiationless processes. One of them can be intersystem crossing phenomenon induced by a spin-orbit coupling enhancement produced by a heavy atom. Additionally, it is known that in long-wavelength emitters, enhanced internal conversion is responsible for quenching emissive states.[68]

Did they show any quenching effect with different polarity of solvents?

According to emission quantum yield values, presented in table 3 in manuscript the polarity of solvent doesn’t affect this parameter. The small polarity effect was observed for molecule C4 in which the PLQY drops with increasing the polarity of solvent.

(Figure 6) PL spectra on C1 and C5 show side emission peaks, what is the reason behind that?

Following the recommendation of the referee we have added some explanation of the observed behavior. The added text is as follows:

“These results, together with the Stokes shift, much smaller in the case of C1 and C5 molecules, may suggest more rigid structure and smaller geometry and electronic density distribution changes during the excitation in these two molecules.”

Authors must add more discussion about D-π-A-π-D types molecules and compare the device performances with current study. Discussion of device efficiencies while comparing with other reported molecules are very import in case of OLED research. Because readers will get clear idea about molecular design of highly efficient materials when authors provide comparison with similar reported molecules.

In order to answer to these questions (16, 17) and following the recommendation of the referee we have modified and added additional discussion to OLEDs part.

It is obvious that obtained efficiency of OLEDs is so low even for fluorescence OLEDs, that it is no need to compare it with another published data. The above is quite well known and for the purpose of this paper we believe it is not necessary to report these basic facts.

Reviewer 3 Report

  The manuscript is an extension of series of the recent research works on donor-acceptor type organic OLED materials by the authors. The authors investigated pi-conjugated carbazol-based donor-acceptor-donor type molecules (C1 - C5) for OLED. The molecules showed fluorescence in solutions and in spin-coated films and electroluminescence in the model devices. The emission wavelength were controlled by changing the acceptor moiety, and the photophysical properties were explained well by the redox potential (HOMO-LUMO levels) of the molecules. The information presented in this study is considered to be significant. The presentation of the manuscript feels good, and the supplementary information includes many scientific data useful to readers. The manuscript is acceptable for publishing in “Nanomaterials”.

Minor comments:

Some duplicated description: for example, L399, “intermolecular interaction between molecules”, and so on. L502-505, the paragraph should be in “Conclusions”. L372: The spectra of “red complexes”... (the words are only there.) Compound C4 showed NIR luminescence (it is very good). But the quantum yield is very small. The description on the mechanism for this and on the potential of improvement by molecular assembly may be need (it is helpful for readers). The PL for C1 - C4 showed double-exponential decays. The reason for this should be explained. In the supplementary information, the absorption spectra of these molecules were scarcely influenced by solvents, on the other hand, the fluorescence spectra were red-shifted by matrices. The explanation may be needed. “Abstract” and “Conclusion” can be refined well.

Author Response

Minor comments:

Some duplicated description: for example, L399, “intermolecular interaction between molecules”, and so on. L502-505, the paragraph should be in “Conclusions”.

These errors have been amended and discussion about OLEDs and conclusions were revised.

L372: The spectra of “red complexes”... (the words are only there.)

This error has been amended.

Compound C4 showed NIR luminescence (it is very good). But the quantum yield is very small. The description on the mechanism for this and on the potential of improvement by molecular assembly may be need (it is helpful for readers).

Following the recommendation of the referee we have added a tentative explanation of the small quantum yield of C4. The added text is as follows:

“It can be seen that the radiative rate constant, obtained from the PLQY/ τ ratio, systematically decreased upon passing from C5, C1, C3, C2, to C4. Evidently, the introduction of a selenium atom in the acceptor unit decreased the radiative rate constant. Such behavior may be interpreted in terms of relative twisting of the donor and acceptor moieties in the excited state. It is generally known that the radiative rate constant is a function of the twist angle (Θ) and decreases when the value of Θ changes from 0° to 90°, where the two moieties involved in the charge transfer are orbitally decoupled and, as a consequence, the transition is forbidden and kr becomes very small. However, additional experiments and/or calculations are needed to confirm this hypothesis. On the other hand, the very low emission quantum yield observed for C4 can be related to activation of some radiationless processes. One of them can be intersystem crossing phenomenon induced by a spin-orbit coupling enhancement produced by a heavy atom. Additionally, it is known that in long-wavelength emitters, enhanced internal conversion is responsible for quenching emissive states.[68]

 The PL for C1 - C4 showed double-exponential decays. The reason for this should be explained.

In order to answer this question the text in the manuscript was modified:

It should be noticed that for compounds C1, C2, C3, and C4, it was difficult to fit the emission decay with a monoexponential function, where very small contribution of the second time was detected. However the origin of it is unclear at this stage of research. However, in the case of the compound C5, emission decayed monoexponentially (for details, see Fig. S18 and Table S1).

In the supplementary information, the absorption spectra of these molecules were scarcely influenced by solvents, on the other hand, the fluorescence spectra were red-shifted by matrices. The explanation may be needed.

As far as we properly understood this comment:

The influence of polarity on absorption spectra has been commented and added in manuscript:

“Therefore, the absorption and emission spectra of studied compounds were obtained in solvents with different polarities, and the absorption spectra remained unchanged (Figure S16), what may suggest that polarity doesn’t affect the energy of ground states.”

Concerning the fluorescence red shift in matrix (green exciplex PVK:PO-T2T): we believe that explanation was given in manuscript:

“Interestingly, as the dopant concentration increased in the exciplex, a bathochromic shift was observed in the PL spectra (Fig. S20). Additionally, the photoluminescence quantum yields of all thin films were similar and were in the range 4-5%, which suggests strong concentration-induced quenching in the solid state.”

 “Abstract” and “Conclusions” can be refined well.

Following this recommendation from the referee we significantly modified the Abstract and Conclusions.

Round 2

Reviewer 2 Report

Accept